# An Immobilized Form of a Blend of Essential Oils Improves the Density of Beneficial Bacteria, in Addition to Suppressing Pathogens in the Gut and Also Improves the Performance of Chicken Breeding

**DOI:** 10.3390/microorganisms11081960

**Published:** 2023-07-31

**Authors:** Shyam Sundar Paul, Savaram Venkata Rama Rao, Rudra Nath Chatterjee, Mantena Venkata Lakshmi Narasimha Raju, Ajay Kumar Mahato, Bhukya Prakash, Satya Pal Yadav, Alagarsamy Kannan, Godumagadda Narender Reddy, Vikas Kumar, Prakki Santosh Phani Kumar

**Affiliations:** 1Directorate of Poultry Research, Poultry Nutrition, Indian Council of Agricultural Research (ICAR), Hyderabad 500030, India; svramarao1@gmail.com (S.V.R.R.); rnch65@gmail.com (R.N.C.); mvlnraju@gmail.com (M.V.L.N.R.); drbhukyaprakash@gmail.com (B.P.); yadav.satyapal@gmail.com (S.P.Y.); akanna72@gmail.com (A.K.); reddynarender7@gmail.com (G.N.R.); vikassugandhi@gmail.com (V.K.); santosh.prakki@gmail.com (P.S.P.K.); 2The Centre for DNA Fingerprinting and Diagnostics, Department of Biotechnology, Hyderabad 500039, India; akmahato@cdfd.org.in

**Keywords:** gut microbiome, antimicrobial resistance, antibiotic growth promoter, broiler chicken, essential oil, alternatives, amplicon sequencing

## Abstract

Antimicrobial growth promoters (AGP) are used in chicken production to suppress pathogens in the gut and improve performance, but such products tend to suppress beneficial bacteria while favoring the development and spread of antimicrobial resistance. A green alternative to AGP with the ability to suppress pathogens, but with an additional ability to spare beneficial gut bacteria and improve breeding performance is urgently required. We investigated the effect of supplementation of a blend of select essential oils (cinnamon oil, carvacrol, and thyme oil, henceforth referred to as EO; at two doses: 200 g/t and 400 g/t feed) exhibiting an ability to spare Lactobacillus while exhibiting strong *E. coli* inhibition ability under in vitro tests and immobilized in a sunflower oil and calcium alginate matrix, to broiler chickens and compared the effects with those of a probiotic yeast (Y), an AGP virginiamycin (V), and a negative control (C). qPCR analysis of metagenomic DNA from the gut content of experimental chickens indicated a significantly (*p* < 0.05) lower density of *E. coli* in the EO groups as compared to other groups. Amplicon sequence data of the gut microbiome indicated that all the additives had specific significant effects (DESeq2) on the gut microbiome, such as enrichment of uncultured Clostridia in the V and Y groups and uncultured Ruminococcaceae in the EO groups, as compared to the control. LEfSe analysis of the sequence data indicated a high abundance of beneficial bacteria Ruminococcaceae in the EO groups, Faecalibacterium in the Y group, and Blautia in the V group. Supplementation of the immobilized EO at the dose rate of 400 g/ton feed improved body weight gain (by 64 g/bird), feed efficiency (by 5 points), and cellular immunity (skin thickness response to phytoheamagglutinin lectin from *Phaseolus vulgaris* by 58%) significantly (*p* < 0.05), whereas neither yeast nor virginiamycin showed a significant effect on performance parameters. Expression of genes associated with gut barrier and immunity function such as *CLAUDIN1*, *IL6*, *IFNG*, *TLR2A*, and *NOD1* were significantly higher in the EO groups. This study showed that the encapsulated EO mixture can improve the density of beneficial microbes in the gut significantly, with concomitant suppression of potential pathogens such as *E.coli* and improved performance and immunity, and hence, has a high potential to be used as an effective alternative to AGP in poultry.

## 1. Introduction

Optimal gut health in fast-growing birds such as broiler chickens is of paramount importance for achieving their genetic potential and reducing pathogen contamination in poultry products. There is increasing societal concern about the application of antibiotics in feed as growth promoters in chickens in order to improve gut health and feed efficiency, due to their contribution to the development of antimicrobial resistance (AMR) in food animals and dissemination to humans or the environment, as well as the risk of drug residues in chicken products. It has been shown that the use of antibiotics in the feed may favor the selection of resistance to more than one class of antimicrobials in the bacterial community residing in the gut of chickens [1]. Consequently, the development of new green additives as an alternative to antibiotic growth promoters (AGP) has emerged as a priority area in the feed industry and animal production [2]. It has been suggested that AGP most likely works as growth promoter by inhibiting inflammatory responses by intestinal inflammatory cells, which in turn causes changes in the gut microflora [3]. Essential oils represent a green alternative to AGP because they can improve growth, feed efficiency and gut health by different mechanisms. These compounds may modulate the gut microbiome through bacteriostatic/bacteriocidal action, direct effects on host gut development or function, modulation of the immune system, or by the reduction of gut inflammation by regulating release of inflammatory cytokines [4]. Some studies have indicated that feed efficiency and gut health are positively correlated with a population of beneficial gut microbes [5]. Essential oils with pathogen suppression ability (low minimum inhibitory concentration (MIC) for pathogens) and with the ability to spare beneficial microbes (high MIC for beneficial microbes) in the gut are likely to be more effective as gut toning additives than those that affect both groups of microbes [5]. EOs are volatile and can evaporate quickly during feed processing or may become metabolized by being absorbed in the stomach or foregut before reaching their main site of action, i.e., the intestine and hindgut. Immobilization or entrapment in an organic matrix such as calcium alginate microspheres can reduce the loss of essential oils by evaporation, reduce effective dose, delay the absorption of these molecules, and allow the progressive release of the essential oils at the hindgut or intestine [6,7]. A limited section of the literature has focused on the effects of the use of immobilized beneficial bacteria-sparing essential oils on gut health, immune function, and microbiome in broiler chickens.

The objective of this study was comprehensively to evaluate the effect of an immobilized blend of essential oils having beneficial bacteria-sparing ability on breeding performance, intestinal health and function, and the gut microbiomes of broiler chickens, and to analyze the feasibility of replacing antibiotics in chicken farms, including understanding the mechanisms of the actions of essential oils.

## 2. Materials and Methods

### 2.1. Feed Additives

#### 2.1.1. Preparation of Immobilized Essential Oil Microspheres

In the present study, a composite feed additive developed in our lab in an earlier study, and containing cinnamon oil, thyme oil, and carvacrol immobilized in a matrix of calcium alginate and sunflower oil for targeted delivery of the EO blend in the intestines of chickens, was utilized to study its effects on broiler chickens [8]. The three plant bioactive substances (essential oils or their components), namely, cinnamon oil, thyme oil, and carvacrol were found to have the strongest *Escherichia coli* inhibiting activity, while sparing *Lactobacillus* spp., among nine essential oils or their components tested in a series of in vitro culture experiments in one of our earlier studies. All the three essential oils (cinnamon oil, Ceylon type FCC, FG, material number W229202-1KG-K; carvacrol, natural 99% FG, material number W224511-100G-K; and thyme oil, white FG, material number W306509-1 KG-K) were purchased from Sigma-Aldrich Co., 3050 Spruce Street, St. Louis, MO 63103, USA, via its New Delhi office. A 30 mL quantum of 2% (*w*/*v*) solution of sodium alginate was prepared by adding 0.6 g sodium algenate to 30 mL of distilled water. Next, 1 mL cinnamon oil, 1 mL thyme oil, 1 mL carvacrol, and 3 mL of sunflower oil were added to the algenate solution and mixed well by stirring at ambient temperature. The mixture was taken into a syringe and extruded through a 16-gauge needle, drop by drop, into a 100 mL solution of 0.1 M calcium chloride solution and allowed to remain there for 30 min for hardening. Calcium chloride was decanted, and the microsphere beads were rinsed two times in distilled water, and then air dried under a fan in ambient temperature by being placed on a paper towel. After drying, they were stored in air-tight amber colored bottles until used. The weights of the beads were recorded. Diameters of the dried beads were also measured, using a Vernier caliper. Recovery of each EO component of the plant bioactive substances was tested by preparing beads using each component, followed by dissolving the beads and assay of each component using a standard spectrophotometric method. Similarly, recovery of sunflower oil was also tested by solvent extraction. The average recovery of plant bioactive substances and sunflower oil was above 80%. The concentrations of the components of the dried microspheres were approximately (*w*/*w*): calcium alginate 9–10%, sunflower oil 45%, cinnamon oil 14–15%, carvacrol 14–15% and thyme oil 14–15%.

The air-dried microspheres (0.85–1.25 mm diameter) of the composite additive (EO) were found to have been dissolved completely in gastric pH within 5 min on shaking, but remained intact at a pH range of 6–8. Essential oil microspheres were stored in amber colored air-tight glass jars until used.

#### 2.1.2. Other Additives

A commercial live yeast *(Saccharomyces cerevisiae*; 20 billion colonies forming units/g) was also used in one experimental group at the dose level recommended by the manufacturer (500 g/ton feed).

An antibiotic growth promoter, namely, virginiamycin, was also used as a positive control in one experimental group at the dose level recommended by its manufacturer (40 g/ton feed).

### 2.2. Animals, Treatments, and Experimental Design

A total of 300 newly hatched chicks obtained from a commercial hatchery (Venkateswara Hatcheries Pvt Ltd., Hyderabad 500001, India) were randomly divided into 60 pens measuring 6 ft^2^ (stainless steel battery brooder cages), containing 5 birds in each pen. Twelve replicated pens were allotted to each of the 5 groups in a completely randomized design. All of the groups were offered the same basal diet. The negative control (C) group’s feed was not supplemented with any AGP or alternatives to AGP feed additive. The four other groups were supplemented with one of the five additives, namely, virginiamycin (V, 40 g/t feed; as recommended by the manufacturer), yeast (Y, 500 g/t feed; as per the dosage recommended by the manufacturer), EO1 (EO, 200 g/t feed), and EO2 (EO, 400 g/t feed).

All of the chickens were fed their diets ad libitum as per the feeding standards recommended by the supplier of the chicken strain. A detailed description of diet composition has been presented in Appendix A. The institute’s (ICAR-DPR) animal ethics committee approved the experiments (sanction number IAEC/DPR/18/3), and all of the animal handling and sampling procedures were performed, as per the guidelines of the animal ethics committee. The birds were housed in an open-sided poultry house. The range of minimum and maximum temperatures in the house during the experiments was 23.5 to 35.5 °C. All of the birds were wing-tagged, weighed, and vaccinated with Marek’s disease vaccine on arrival. Brooding was performed at a temperature range of 30–33 °C for up to 21 days, with the help of incandescent bulbs. The feed was offered and made available in the feeders placed in each pen. A weighed quantity of feed was offered daily and leftover feed was weighted at weekly intervals. The feed for each phase was prepared in a single batch and stored in a container with a lid for daily use. All of the birds were provided with ad libitum access to the drinking water through separate drinkers in each pen. Body weight (BW) was recorded at weekly intervals for each pen and a per bird average weight was calculated for each pen. The birds were vaccinated with Newcastle disease (Lasota strain) vaccine on the 5th and 28th day and with infectious bursal disease vaccine on the 10th and 16th day. Mortality was recorded daily, and the pen number, the wing band number, and the body weight of the dead bird were recorded. The feed conversion ratio (FCR) was adjusted for mortality. Cages were placed distant from each other to avoid fecal contamination between pens.

### 2.3. Metabolism Trial

A metabolism trial was conducted on birds during the 4th week of age, involving four replicate pens from each group (20 birds per group), and using the total collection method. Briefly, total excreta were collected, weighed, and the dry matter (DM) content was estimated in each pen for 3 consecutive days, and the samples were then pooled pen-wise for analysis. The daily feed intake in each pen was also recorded. The nitrogen retention and apparent digestibility coefficients of DM and EE of feed were calculated according to the formulae for the total collection method [9].

### 2.4. Chemical Analysis of Proximate Principles

The CP (4.2.03; by Kjeldahl method after acid hydrolysis), fat (4.8.01; after extraction with petroleum ether by the Soxhlet method), and ash (4.8.03; by igniting at 550 °C for 3 h in a muffle furnace) contents of feed and excreta were determined using AOAC [10] procedures. The Ca levels were measured using an atomic absorption spectrophotometer according to the methods suggested by its manufacturer (AAnalyst 400, PerkinElmer, and Shelton, CT, USA). The P levels were estimated using a colorimetric procedure [11].

### 2.5. Immunity Parameters

#### 2.5.1. Hemagglutination Inhibition (HI) Assay

The HI test was carried out on serum samples (10/group) collected at the 5th week of age. The HI titer of the Newcastle disease virus (NDV) antigen (LaSota virus stock) was adjusted by dilution to contain 4 units of hemagglutination activity. The HI titer was determined as being the highest dilution of serum samples that inhibited agglutination of chicken RBCs by NDV.

#### 2.5.2. Cutaneous Basophil Hypersensitivity Test

The cell-mediated immune (CMI) response was assessed during the 5th week of age by a cutaneous basophilic hypersensitivity test using phytohemagglutinin lectin from *Phaseolus vulgaris* (PHAP). Birds (6/group) were injected intra-dermally on the leg toe web with 0.1 mg PHAP in 0.1 mL PBS. The thickness was measured using a thickness gauge before and 24 h after injecting mitogen. The CMI response was observed by the difference in the thickness before and after injecting with PHAP.

#### 2.5.3. Collection of Blood Samples and Biochemical Analysis

During experiment II, blood samples were taken from the brachial veins of 10 birds/group on d 28, and then placed in Vacutainer tubes and utilized for the separation of serum. Approximately 8 mL of blood was collected from randomly selected birds using a 21-gauge needle. Serum was separated, stored at −20 °C and analyzed for glutathione reductase [12], glutathione peroxidase [13], superoxide dismutase [14], and lipid peroxidation [15].

#### 2.5.4. Sample Collection, RNA Extraction, and mRNA Expression Analysis

Gene expression analysis in the intestinal tissue (from jejunum; approximately 1 cm) of the chickens was performed for four gene groups, as described earlier [16]. Briefly, intestinal tissues were utilized from 6 birds in each of the groups. Total RNA was extracted using Trizol reagent (15596018; Invitrogen ThermoFisher Scientific, Carlsbad, CA, USA), according to the manufacturer’s protocol. The concentration and purity of the RNA were analyzed using a NanoDrop 2000 spectrophotometer (Thermo Scientific, Waltham, MA, USA). The quality of the total RNA was checked using agarose gel electrophoresis.

One μg of total RNA (per sample) was reverse transcribed into complementary DNA (cDNA) using a RevertAid first-strand cDNA synthesis kit (K1622; ThermoFisher Scientific Baltics, Vilnius, Lithuania) as per the manufacturer’s instruction manual. Real-time PCR was performed in triplicate with suitable controls using the Maxima SYBR Green/ROX qPCR master mix (K0221; ThermoFisher Scientific, Waltham, MA, USA) and an ABI StepOne quantitative PCR thermal cycler, by following the manufacturers’ guidelines. Details of quantitative real-time PCR primers and procedures used in this study were the same as described earlier [11]. qPCR products were checked through agarose gel electrophoresis for the presence of a single band/product. Two reference (housekeeping) genes, namely, glyceraldehydes-3-phosphate dehydrogenase (GAPDH) and beta-actin, were also subjected to qPCR assay in each sample.

Relative gene expression levels of each target were calculated using the 2^−ΔΔCt^ method [17]. Normalization of the expression levels of the target genes was performed with the geometric mean of two housekeeping genes, glucose-6-phosphate dehydrogenase (*GAPDH*) and beta-actin (*ACTB*), as described earlier [18].

#### 2.5.5. Gut Content Sample Collection

On the 43rd day of age, ten healthy chickens were selected (one per pen) per group, caught, and euthanized by cervical dislocation.

The gut was opened using sterile scissors, and the luminal contents of the hindgut (from the duodenum to the cloaca, including caeca) were collected into sterile storage vials. For every g of gut content, 5 mL 1X phosphate buffer saline was added and mixed by vortexing to produce a homogenate. The homogenized gut content was immediately stored in a freezer at −20 °C, transported to the laboratory, and stored at −80 °C.

### 2.6. DNA Extraction

DNA was extracted from the homogenized hindgut contents of individual chickens following the bead beating plus column method described by Yu and Morrison [19] using the DNA purification columns from the QIAamp Fast DNA Stool Mini kit (QIAGEN GmbH, Hilden, Germany). DNA concentration and quality were assessed using a Qubit 3.0 fluorometer (ThermoFisher Scientific, Waltham, MA, USA; #Q33238), a DNA HS Assay Kit (ThermoFisher Scientific, Walthan, MA, USA; #Q32851), and also by gel electrophoresis. DNA was stored in a freezer at −20 °C until further processing.

### 2.7. Amplicon Sequencing and Analysis

The hindgut microbiome was characterized by sequencing the v3–v4 region of the 16S rRNA gene for 30 DNA samples (from 6 birds per group slaughtered at 43 d of age). Extracted DNA was PCR-amplified using the primer pair S-D-Bact-0342-b-S-17 (5′-CCTACGGGNGGCWGCAG-3′) and S-D-Bact-0785-1-A-21 (5′-GACTACHVGGGTATCTAATCC-3′), as recommended by Klindworth et al. [20], and as per the protocol described earlier [1]. The amplified product was checked on agarose gel and gel purification was performed to remove non-specific PCR products. In total, 5 ng of amplified product was used for library preparation using the NEBNext Ultra DNA library preparation kit (New England Biolabs, Ipswich, MA, USA). The quantification and quality estimation of the library was performed in an Agilent 2200 Tape Station. Sequencing was performed using an Illumina HiSeq sequencer in 2 × 250 bp pair–end sequencing mode. Three negative controls per plate were also placed in the sequencing run to detect spurious amplifications, if any. Analysis of the sequence data largely was carried out as described earlier [1,21]. The raw reads were demultiplexed and filtered using FastQC (version 0.11.8). Primers and adapters were trimmed using in-house Perl scripts. The MOTHUR software package (V 1.47.0) [22] was utilized for further quality screening, assembly of reads, alignment against SILVA [23] seed alignment (v138), trimming of alignment, denoising using pre-cluster algorithm, chimera removal using the UCHIME algorithm [24], and taxonomic classification using the RDP classifier [25]. Sequences classified as unrelated taxons (other than bacteria) were removed. Clean sequences were subjected to operational taxonomic unit (OTU) clustering using DMSC software [26] at a 97% similarity cutoff. DMSC output was converted to a MOTHUR-formatted list file and a shared file by using various shell commands. Consensus taxonomy files for the OTUs were created using the RDP classifier algorithm in MOTHUR, using the Silva database, version 138.1 [23]. MOTHUR-formatted shared files and consensus taxonomy files were further analyzed using MicrobiomeAnalyst [27] web server for the analysis of abundance bar plots, alpha diversity, beta diversity, rarefaction curve, differential abundance, and biomarker identification. For the analysis of beta diversity (principal coordinate analysis (PCoA) and nonmetric multidimensional scaling (NMDS)), data were normalized by the cumulative sum scaling (CSS) method after disabling the default data filtering options for low counts and low variances. For the analysis of alpha diversity, data were rarefied to the minimum library size (at 8835 sequences per sample). For the biomarker identification (using linear discriminant analysis effect size or LEfSe) and differential abundance (using DeSeq2), data were normalized using the CSS option and filtered for low counts and low variances using the default setting in order to focus on important features. Beta diversity analyses were carried out at different taxonomic levels using PCoA as well as NMDS ordination, based on different distance methods, and using permutational multivariate analysis of variance (PERMANOVA) and homogeneity of group dispersion (PERMDISP) tools. Differential abundance was analyzed using DeSeq2 followed by Benjamini–Hochberg false discovery rate (FDR) correction for multiple comparisons for detection of significant differences. Groups were compared pair-wise using the nonparametric Mann–Whitney U test (Wilcoxon rank sum test) as implemented in SPSS [28]. Group-specific biomarkers were analyzed using the LEfSe algorithm at an FDR adjusted *p*-value cutoff value of 0.05 and logarithmic LDA score cutoffs of 2 and 3.5 [24]. The LEfSe bar plot at the genus level was created using the MicrobiomeAnalyst web server, utilizing an LDA score cutoff of 2.0, but the LEfSe cladogram was created using a standalone version of LEfSe [29], using an LDA score cutoff of 3.5. Data were rarefied to the minimum library size (at 8835 sequences per sample) without data filtering for rare OTUs for rarefaction analyses. Bubble plots and UpSet diagrams were created using the R package. For the creation of the upset diagrams, data were filtered for the low count and low variance OTUs (OTUs with <5 members or appearing in <2 samples were removed) to focus on major OTUs only.

### 2.8. qPCR Analyses

#### Quantification of Population Sizes of *Escherichia coli* and *Lactobacillus* spp.

The density of *E. coli*, *Lactobacillus* spp., and total bacteria were estimated (in 6 replicate samples per group) using a Maxima SYBR-Green-based quantitative real time PCR (qPCR) master mix (Genetix Biotech Asia Pvt Ltd., New Delhi, India) and an ABI StepOne quantitative PCR thermal cycler (ThermoFisher Scientific, MA, USA), using respective specific primers [25,26]. qPCR reaction was performed in triplicate, along with controls, as per the qPCR master-mix manufacturer’s instruction. Sample-derived qPCR standards were prepared and utilized as described earlier [30]. The absolute abundances were expressed as the number of 16S rRNA gene copies/50 ng DNA samples.

### 2.9. Statistical Analysis

The data on BWG, FCR, slaughter parameters, digestibility, and qPCR data on *E. coli*, *Lactobacillus* spp. and total bacteria were analyzed by one-way analysis of variance using SPSS (2008). Specific differences between pairs of means were tested using Duncan’s multiple range test at *p* < 0.05. All expression data were tested for equal variance (Levane’s test) and normality (Shapiro–Wilk test). For statistical analysis, normalized gene expression data were square-root-transformed [31]. Expression data were analyzed by using the nonparametric Kruskal–Wallis test followed by the Dunn post hoc test and FDR adjustment of *p*-values. All expression data presented in the Table are the geometric mean of untransformed relative mRNA levels [32]. The stability of reference genes used for the normalization was checked using NormFinder [33].

For 16S amplicon sequencing data, on detection of significant difference in overall abundance between groups on DESeq2 analysis, followed by FDR correction of *p* values, groups were compared using the Wilcoxon rank-sum test as implemented in SPSS [28]. Alpha diversity metrics were analyzed using the nonparametric Wilcoxon test. For beta diversity metrics, significance testing was carried out using PERMANOVA and PERMDISP.

## 3. Results

### 3.1. Effects on Performance Parameters and Digestibility

During the first two weeks, FE was significantly better in the V group as compared to other groups (Table 1).

However, at the end of the 5th week, cumulative BWG was significantly lower in the V group as compared to other groups, but FE was comparable among all the groups. In the 6th week, cumulative BWG was significantly higher in the EO2 group as compared to other groups, and the V group had the lowest BWG. In the 6th week, FE was best in the EO1 and EO2 groups, significantly better than the V group, and also about 5 points better than the negative control group. FE and BWG were not significantly influenced by the supplementation of yeast.

Overall, there were 3.09, 2.39, 2.31, 1.97, and 1.88% cumulative average mortality rates in the groups C, V, Y, EO1, and EO2, respectively, and mortality was random between pens, with no significant treatment effect.

There was no significant difference in the digestibility of ether extract (EE), crude protein (CP), or dry matter (DM) among the groups; however, a trend in improvement in fat digestion with the inclusion of Y, EO1, and EO2 was evident (Appendix A).

### 3.2. Effects on Humoral and Cellular Immune Response and Antioxidant Function

In the present study, cellular immunity response in terms of PHAP response was significantly higher in both EO groups, as compared to control and other groups, but humoral immunity in terms of HI titer against NDV was not significantly influenced by the treatments (Appendix A). Two serum antioxidant enzymes, namely glutathione reductase and superoxide dismutase, were not influenced by the treatments, but glutathione peroxidase levels were significantly lower in both of the EO groups as compared to the control and the other groups. There was no difference in the levels of lipid peroxidation between the groups, indicating that there was no significant change in oxidative stress in tissues due to the treatments.

### 3.3. Gene Sxpression in Chicken Gut Tissue

The relative normalized expression of barrier-forming gene claudin 1 was significantly higher in EO groups and Y than in the negative control and the virginiamycin group (Table 2).

Relative normalized expression of pattern recognition receptor genes such as toll-like receptor 2A and nucleotide-binding oligomerization domain 1 was also significantly higher in the EO groups and Y than in the negative control and the virginiamycin group. Relative normalized expression of cytokine IL6 was also significantly higher in the EO groups and Y than that of the negative control and the virginiamycin group. Expression of IFNG was significantly higher in all the treatment groups as compared to the negative control, with highest values in the EO200 and Y groups.

### 3.4. Effects on the Density of E. coli and Lactobacillus in Gut Content

The qPCR assay indicated that the EO200 and EO400 groups had a significantly lower density of *E. coli* as compared to the control, Y, or V groups (Appendix A). The abundance of *Lactobacillus* spp. was also lower in the EO200 and EO400 groups as compared to the control, Y, and V groups. The density of *Lactobacillus* spp. was significantly higher in the EO400 group as compared to the EO200 group. However, the density of total bacteria was not significantly influenced by the treatments.

### 3.5. Microbiome Sequencing

The 16S amplicon sequencing generated 4,129,358 raw reads with 885,343 quality passed (after merging, quality screening, pre-clustering, and chimera removal) consensus sequences or contigs (range: 8835 to 62,571 per sample; average: 29,511 per sample).

### 3.6. Operational Taxonomic Unit (OTU) Abundance

A total of 82,278 OTUs were identified. From 82,278 OTUs, 66,899 OTUs with less than two members or occurring only once were removed, and the remaining 15,379 OTUs were selected for further analysis in order to focus on important OTUs only, remove potentially spurious OTUs, and improve the downstream statistical analysis.

### 3.7. Taxonomy Assignment

At the class level, Clostridia was the most dominant class in all the groups (Figure 1).

The proportion of Bacteroidia was lower in the control group, higher in EO2 and Y groups, and intermediate in the V group. Contrastingly, the proportion of actinobacteria was higher in the V group as compared to the remaining groups including the control. The average abundance of unclassified firmicutes was higher in the EO1 and EO2 groups as compared to others, including the control. The average abundance of unclassified bacteria was higher in the control group as compared to other groups.

The UpSet diagram indicated a high level of overlap of non-rare OTUs (OTUs with at least five members and occurring in at least two samples) among different groups. A total of 357 OTUs were detected in all the groups (Figure 2).

A total of 279 OTUs not detected in the control group were detected in all of the remaining groups. A total of 19 OTUs present in the control group were not detected in any of the remaining groups. A total of 13 OTUs were detected in Y, EO1, and EO2, but not in the control or the V group.

### 3.8. Alpha Diversity and Rarefaction Analysis

Among alpha diversity metrics, Shannon and Simpson’s index did not differ significantly among the groups, whereas the Fisher index was significantly different among the groups. Richness estimators such as observed richness, Chao1, and ACE also differed significantly among groups. The Chao1 index was significantly higher in the EO2 group and tended to be higher in the EO1 and Y groups as compared to the control. The Fisher index was significantly higher in the EO2 group as compared to the control (Appendix A).

The rarefaction curve indicates the relation between the number of sequences and the number of taxonomic OTUs detected, and the steeper the slope, the higher the diversity. The rarefaction curve (Appendix A) approached the asymptotic level for each group, indicating the availability of sufficient reads to represent each microbial community.

### 3.9. Microbial Beta Diversity

The beta diversity metrics, or ordination plots, depict the partitioning of biological diversity among groups along a gradient, i.e., the number of species shared between two groups.

Beta diversity was visualized using nMDS as well as PCoA ordination methods, but due to space limitations, only two plots, both obtained through NMDS ordination, are presented (Figure 3).

Beta diversity ordination using PCoA or NMDS plots resulted in the clear visual separation of samples due to groups at both the OTU and phylum levels. There was a high degree of overlap among the treatment groups and very little overlap between the treatment groups and the control group. PERMANOVA tests performed using all beta diversity metrics used in this study showed significant (*p* < 0.01) differences in community structure among the different groups (Appendix A). At the OTU level, Jensen–Shannon-based PERMANOVA analysis had the highest Pseudo–F (5.613) and R^2^ (0.473) values among all distance metrics, indicating that 47.3% of microbiota variation is explained by this category (group), in addition to a significant *p*-value (*p* < 0.01). At the phylum level, Jensen–Shannon-based PERMANOVA analysis also had the highest Pseudo–F (5.284) and R^2^ (0.458) values among all distance metrics, indicating that 45.8% of microbiota variation is explained by this category (group), in addition to a significant *p*-value (*p* < 0.01). PERMANOVA tests performed using all beta diversity metrics used in this study showed significant (*p* < 0.01) differences in community structure among different groups at both the OTU and phylum levels.

The beta dispersion values (PERMDISP) were non-significant for all groups in all diversity metrics, both at the OTU and phylum level, indicating a homogeneous dispersion among groups.

### 3.10. Differential Abundances of Bacteria

Differential abundance analysis using DESeq2 on CSS normalized amplicon sequencing data (OTUs with at least five members) followed by FDR correction indicated that three phyla, four classes, twelve families, thirty-four genera, and four hundred twenty-three OTUs were significantly different in abundance among the groups. At the class level, Bacteroidia, Alphaproteobacteria, Epsilonproteobacteria, and 4C0d_2 differed significantly in abundance among the groups. The abundance of about 16 genera, including Bacteroides, Parabacteroides, Butyricimonas, and Faecalibacterium, were significantly lower in C but comparable among the other groups. The abundance of genera such as Dorea, Eggerthella, and Clostridium were significantly higher in the C group than in the others. The abundance of the genus-level uncultured group under Ruminococcaceae and YS2_U was significantly higher in the EO1 and EO2 groups. The abundance of the genus-level uncultured group under class Clostridia was significantly higher in the V and Y groups as compared to other groups, including the C group. The abundance of the genus Clostridium was lowest in the V group and intermediate in the EO1 and EO2 group, but significantly higher in the control group.

DESeq2 analysis of CSS normalized sequencing with FDR correction indicated that 423 OTUs out of 916 major OTUs (having at least five members) were significantly different in abundance levels among the groups (Figure 4).

Pairwise comparison among groups using the Mann–Whitney U test indicated that there were significant differences in the abundance of many OTUs between different groups. Interestingly, the abundance of OTUs, such as OTU1000062 (order Bacteroidales), OTU104145 (family Rikenellaceae), OTU107044 (genus Allistipes), OTU1104680 (family Barnesiellaceae), OTU4325509 (species Robinsoniela peoriensis), OTU4339144 (genus Butyricimonas), OTU586453 (family Christensellaceae) and OTU689975 (genus Parabacteroides), was higher in all the treatment groups than in the control group. The abundance of OTU1010876 (genus Oscillospira) was higher in the EO2 group than in the control or the V group. The abundance of OTU361186 (genus Blautia) was lower in the EO2 and control groups as compared to the other groups. The abundance of OTU100567 (genus Ruminococcus) and OTU839684 (family Lachnospiraceae) were comparable among all the groups.

### 3.11. The Group-Specific Biomarkers Based on the LEfSe Algorithm

The LEfSe analysis identified biomarkers in the gut microbiota (specific taxa that vary in abundance consistently by group) that were indicative of the gut microbiota of each group. A total of 13 taxonomic OTUs were identified as having an LDA score > 2 at the phylotype-OTU level (Figure 5). (Phylotype-OTUs were obtained after merging distance-based OTUs with the same consensus taxonomy.)

A high abundance of genera or genus-equivalent taxonomic groups such as Anaerostipes and an unclassified genus equivalent group under the family Ruminococcacaeae were typical for the control group. A high abundance of Alistipes, unclassified Clostridiales, Parabacteroides, and Faecalibacterium were typical for the Y group. A high abundance of Blautia was typical for the virginiamycin group. A higher abundance of uncultured bacteria, unclassified Firmicutes, and unclassified Bacteroidales were typical of the EO1 group. Similarly, a higher abundance of unclassified Ruminococcaceae, Bacteroides, and Barnesiella was typical of the EO2 group.

A cladogram of important biomarkers identified at different taxonomic levels in different groups, using LEfSe, with an LDA score >3.5 has been presented in Figure 5b. The family Lachnospiraceae was the top biomarker in the control group. The family Christensenellaceae was the top biomarker for the virginiamycin group. The class Bacteroidia, order Bacteroidales, and families, such as Rikenellaceae and Odoribacteraceae were top biomarkers for the Y group. The class Flavobacteriia and order Flavobacteriales were major biomarkers in group EO1. Similarly, the order Rhodospirillales and families, such as Ruminococcaceae, Bacteroidaceae, Marinilabiaceae, and Porphyromonadaceae, were top biomarkers in the EO2 group.

## 4. Discussion

Earlier, we observed that *E.coli* was the most abundant bacteria among known potential pathogenic bacteria and that Lactobacillus was one of the most abundant beneficial bacteria in broiler chicken [21]. Hence, *E.coli* and Lactobacillus were used as representatives of potentially pathogenic and beneficial bacteria, respectively, for selecting EOs via an in vitro study.

In the present study, EO groups showed significantly higher FE and BWG than did the AGP and negative control groups. The improvement in BW and FE was 2.64% and 3.32%, respectively, which amounts to 64 g higher BW/bird and a saving of 137 g feed/bird over the 42 d of the production period, which is of commercial significance. A trend in the improvement of fat digestion with EO1 and EO2 was also observed. Many studies have documented the effects of dietary EOs, individually or in combinations, on the performance of poultry and swine breeding but with varying and conflicting results [33,34,35]. Earlier, Saleh et al. [36] reported that dietary supplementation of thyme and ginger oil individually at dose levels up to 300 g/ton did not affect growth or feed efficiency in broiler chicken. In a review, Zeng et al. [37] summarized results from 12 trials on broiler chicken involving EOs at dose levels ranging from 75 to 1200 g/ton and observed that, on average, the performance improvement was 3 and 3% for weight gain and feed conversion, respectively, whereas many studies have reported a negative effect of EO on growth or feed efficiency. The activities of EO depend on active components and synergistic interactions between components and modes of administration. Although a few studies have reported the effects of combinations of thymol, carvacrol, and cinnamaldehyde on the growth, feed efficiency, and intestinal health of broiler chickens, the effects were not consistent [35,38,39]. Saki et al. [38] reported that thymol plus carvacrol significantly improved BWG and FCR in broilers. Reis et al. [39] used a commercial phytogenic feed additive containing a mixture of essential oils, mainly carvacrol, thymol, and cinnamic aldehyde (exact composition not given), at dose levels of 0.5% and 1% in the birds’ diet and reported that the addition of 0.5% of the additive improved the live weight of supplemental birds significantly compared to the control group at 35 and 42 days of age, while the 1% level group did not improve live weight significantly over the control. Earlier Su et al. [35] reported that a blend of EO (3.05% thymol, 2.3% carvacrol, and 0.26% cinnamaldehyde in dextrin as a carrier) had a significant effect on the expression of gut nutrient transport and barrier function genes, the activity of intestinal enzymes or concentration of SIgA, and nutrient digestibility when used at a dose level of 200 to 400 mg/kg of the additive but that there was no significant effect of the EO supplementation on ADG at 42d of age.

The present study indicated that the supplementation of the immobilized EO blend increased cellular immunity (PHAP response) significantly but had no significant effect on humoral immunity (HI titer against ND virus). In contrast, El-Shall et al. [40] reported that supplementation of a commercial EO mixture (containing 50 g oregano oil, 10 g carvacrol, 33.33 g thyme oil, 50 g eucalyptus oil, 5 g thymol, 10 g eucalyptol, and 27 g acacia gum surfactant in water up to 1 L) in water at a dose of 0.5 mL/L showed an immune-stimulating response to ND and IBD vaccines, and an antiviral effect against ND virus; however, it did not have a growth-promoting effect. The obtained results of cellular immune response in this study were in agreement with those of Acamovic and Brooker [41], who claimed that the polyphenol compound EO and thymol have a stimulating effect on cellular immunity. In this study, the EO mix had no significant effect on lipid peroxidation (MDA) level or SOD or glutathione reductase, which is in agreement with El-Shall et al. [40]. In contrast, Hashemipour et al. [42] reported that thymol, carvacrol, and thyme oil had strong antioxidant activity and inhibited lipid peroxidation. The possible cause of the lower level of glutathione peroxidase in the EO groups as observed in the present study needs further investigation.

The epithelial barrier in the gut plays a vital role in eliminating potential pathogens while maintaining a mutually beneficial relationship with the commensal microbiota. In this study, EO supplementation increased the expression of genes involved in the expression of Claudin-1, a barrier-forming protein of epithelia that helps in maintaining structural integrity and reducing the chance of gut invasion, and our results are in agreement with the reports of Liu et al. [43] and Su et al. [35]. Furthermore, EO supplementation also improved the expression of pattern recognition receptor genes such as TLR2A and NOD1, which detect infection and trigger inflammation to defend against pathogens [16]. IFNG is a type II IFN cytokine critical to both innate and adaptive immunity and which helps fight against viral and bacterial infections [44]. In this study, EO supplementation increased the expression of IFNG significantly.

The inconsistencies in the efficacy of EOs reported earlier may be attributed to potential interactions of EOs and feed components or to EOs’ sensitivity to environmental variables such as light, heat, buffer, etc. Reports on the effects of the use of an immobilized form of EOs in calcium alginate on broiler chicken performance are not available. The current study has demonstrated that the use of the immobilized form of a specific combination of thymol, carvacrol, and cinnamon oil can serve as a potent alternative to antibiotic growth promoter and can significantly improve growth or feed efficiency at a much lower dose than those reported for the conventional route of administration, i.e., directly mixing EOs in feed or water. Recently, Moharreri et al. [45] reported that a mixture of essential oils comprising thyme (50%), summer savory (25%), peppermint (12.5%), and black pepper seed (12.5%) microencapsulated by spray drying in protecting-wall materials (whey protein concentrate, modified starch, and maltodextrin) when used in broiler chickens challenged with S. enteritidis at a dose level of 0.5 to 2 kg/ton feed, improved BW, FCR, body antioxidant status, and modulated intestinal microbial population. In our study, a 0.4 kg/t dose level of the immobilized EO blend was found to be adequate both for BWG and FE response up to 6 weeks of age. Thus, the immobilized EO used in the current study performed more efficiently than did the spray-drying-based encapsulated EO reported earlier, and the effective dose as observed in the current study is much lower than those reported from earlier studies involving direct administration of unprotected EOs in feed.

Several hypotheses have been put forward to explain how AGP or EOs or live yeast may influence host performance or gut health, including limiting opportunistic pathogens and subclinical infections, decreasing microbial competition for host nutrients, modulation of host fat digestion, inhibition of the production of toxins in the gut, regulation of immunity and inflammation in host, and improvement of nitrogen balance [46,47]. However, the gut microbiome is involved in most of these proposed mechanisms, and modulating the gut microbiome has received wide attention in the fight against AMR. Earlier, Lin [48] reported that antibiotic growth promoters may enhance fat digestibility by targeting intestinal bile salt hydrolase producers., but in the current study, the AGP virginiamycin did not increase fat digestibility, and the population density of one of the known bile salt hydrolase-producing bacteria Lactobacillus was not affected by the AGP. On the other hand, the EO groups and Y showed a trend towards improvement in fat digestibility, indicating the possibility of modulation of bile salt hydrolase-producing microbes.

The chicken gut harbors a variety of microbiota having a strong impact on the performance and health of the chicken. The introduction of next-generation sequencing and molecular technologies has allowed for detailed characterization and monitoring of the gut microbiome quickly and robustly. Chicken gut harbors a variety of pathogenic or potentially pathogenic bacteria, such as *E. coli*, *C. perfringens*, *Salmonella* spp., Campylobacter, and Enterococcus, which are known to damage the intestinal epithelium, and which adversely affect the digestion and absorption function of the host and invade the host’s system, causing performance loss and mortality [49], in addition to posing a risk of transmission of zoonotic diseases to humans via chicken meat or egg. On the other hand, there are a variety of beneficial bacteria in the guts of chickens that have probiotic and health-promoting effects, such as improving the intestinal barrier of the host, competitive exclusion of potential pathogens, and maintaining homeostasis in the gut. Some of these beneficial probiotic gut bacteria, such as *Lactobacillus* spp. or *Bacillus* spp., can also produce bacteriocins (a group of antimicrobial peptides) to selectively inhibit the growth of other bacteria, including pathogenic bacteria such as Campylobacter, *Salmonella enteritidis,* and *C. jejuni* [50]. Several newer categories of beneficial bacteria have been shown to be associated with good gut health, such as Lachnospiraceae and Ruminococcaceae, with short-chain fatty acid production and fiber-digestion ability [51,52]; *Faecalibacterium* spp., with the ability to stimulate gut immunity and healthy metabolism [53]; and Blautia, a genus under the Lachnospiraceae family with the ability to alleviate inflammatory diseases and metabolic diseases, and with antibacterial activity against specific microorganisms [54]. A high correlation between the abundance of beneficial microbes in the gut and feed efficiency in chickens has been reported [55]. Hence, a detailed analysis of the change in the population density of beneficial microbes and potentially pathogenic bacteria in response to any feed additive can give detailed insight into the gut-health-promoting potential of such additives.

In the present study, supplementation of broiler diets with the AGP virginiamycin showed no growth-promoting effect except for the 1–14 d period, when FE was significantly better in the V group as compared to other groups. Supplementation with the AGP also did not influence the apparent digestibility of nutrients.

The effects of AGP supplementation on performance response have remained inconsistent. Several studies have shown no weight gain difference in broilers fed an AGP diet, in the absence of health problems [56,57]. However, other studies have reported significant effects of AGP on broiler weight gain or feed efficiency [58,59]. In a meta-analysis involving 174 scientific articles containing 183 experiments on broiler chicken, Cardinal et al. [60] reported that higher weight gain and lower feed conversion ratio were observed in AGP-fed groups during the initial phase (1 to 21 d) and the total period (1 to 42 d) with no difference in the final phase (22 to 42 d). In an earlier study, we observed as well that supplementation of AGP did not exert a growth-promoting effect, except for the 1–21 d period in one of the three experimental feeding trials [1]. A reduction in the effectiveness of AGPs over the last 30 years was suggested by Laxminarayan et al. [61], a reduction which may be partly attributed to the optimization of production conditions and increasing levels of antimicrobial resistance.

Live yeast has been suggested to be one of the most potential probiotics that can be used in animal ration as a potential alternative to feed antibiotics in animals [62], and it has been shown to improve gut integrity and modulate the immune system and microbial communities in birds [63]. However, in the present study, feeding of live yeast did not significantly improve BWG, FE, immunity, or antioxidant parameters, although a trend in improvement of fat digestibility was observed. Earlier, Sousa et al. [64] reported that the inclusion of yeast in the diets of broilers from 22 to 42 d did not affect performance. Similarly, He et al. [46] observed that live yeast supplementation improved ADG or FCR (during d 22–42 but not during d1–21 or across the whole experimental period (1–42 d)), improved CP digestibility, increased ND vaccine antibody titer, and improved serum antioxidant status (increased SOD and decreased MDA level in serum).

The taxonomic assignment data has indicated that individual additives had some characteristic effect on the gut microbiota. Differential abundance analysis using DESeq2 followed by FDR correction has shown significant differences among groups in different taxonomic levels. The taxonomic analysis did not indicate a substantial abundance of *E. coli* or any other known pathogenic bacteria (abundance being < 0.1%), however, qPCR data have indicated that EO200 and EO400 groups had significantly lower *E. coli* density as compared to the control or other groups, while the AGP group had a comparable level of *E. coli* to that of the control. *E. coli* is considered part of normal flora in the gastrointestinal tracts of chickens and is considered one of the most important and frequent pathogens responsible for food-borne diseases in poultry and humans worldwide [65]. Hence, a decrease in *E. coli* in the chicken gut with the use of EO may be important for poultry producers.

Based on the sequencing data, Clostridia under phylum Firmicutes was the most dominant class in all the groups, which is in agreement with our previous report [1]. There was considerable variation between individual birds in the relative abundance of different classes. The average proportion of Bacteroidia was higher in EO2 and Y groups, intermediate in the V group, and lowest in the control group. The UpSet diagram indicated that a considerable number of OTUs present in treatment groups were absent in the control group and that only a few OTUs were unique to each group. In the present study, richness indices were affected by treatments. Several studies have indicated that there are no significant effects of AGPs on alpha diversity metrics [66,67], while others have indicated a decrease [68,69] or an increase [70] in alpha diversity. Kim et al. [63] reported that there was no difference in Chao1 and Shannon indices on live yeast supplementation in broiler chicken. In the present study, supplementation of all the three categories of additives i.e., the AGP, Y, and EO had a significant effect on beta diversity. Several studies have indicated that beta diversity metrics were consistently influenced by AGPs [69,70,71], with only a few studies reporting no significant changes [72]. Kim et al. [63] showed that live yeast significantly influenced beta diversity metrics in chicken gut microbiota. Dietary supplementation of essential oils does not always result in changes to alpha and beta diversity in the microbial populations of poultry GIT [73]. However, the EO mix used in the current study produced increased diversity and evenness. Similar results were also reported by Feye et al. [74]. It has been suggested that the biotransformation of phyotoactives by microbiota contributes to the modulation of the functionality of the intestine, and that such effects are linked to increased diversity and biological activity of the microbial population [74].

Based on DESeq2 analysis, the abundance of 423 OTUs differed significantly among at least one pair of the groups. The most notable effect of supplementation of the additives was differential enrichment of uncultured Clostridia in the V and Y group and uncultured Ruminococcaceae and YS2_U in the EO groups as compared to the control. Similarly, LEfSe analysis identified a high abundance of beneficial bacteria Faecalibacterium in the Y group, Blautia in the V group, and Ruminococcaceae in the EO group. Interestingly, in the current study, amplicon sequencing did not detect a significant difference in the abundance of *Lactobacillus* spp. among the groups. The qPCR assay indicated a decrease in the density of Lactobacillus in the EO groups under in vivo conditions, yet the substantial Lactobacillus sparing activity, as observed in in vitro results, was also achieved under in vivo testing. Moreover, the selected EO mix increased the population of Ruminococcaceae—a known beneficial bacteria. Earlier, dietary supplementation of EO had been shown to increase Lactobacillus populations in the chicken gut [75,76]. It has been reported that live yeast supplementation increased the abundance of the phyla Firmicutes and genera Lactobacillus, Prevotella, and Enterococcus compared with the control group [63].

Collectively, the data presented herein demonstrate that the targeted release of the EO mixture in chicken GIT achieved by supplementing the immobilized EO mix can improve the density of beneficial microbes, significantly improve growth, immunity, and feed efficiency in chicken, and hence, can be used as an effective alternative to antibiotic growth promoter in poultry.

## 5. Conclusions

The present study demonstrated that supplementing the immobilized blend of three EOs, namely, cinnamon oil, thyme oil, and carvacrol, exhibiting strong *E. coli* inhibiting activity and substantial ability to spare beneficial microbes, to chicken can improve ADG, FE, and cellular immunity, reduce the density of *E. coli* and significantly improve the density of beneficial microbes such as Ruminococcaceae at the dose level of 400 g/ton feed, and that the EO-supplemented group performed better than did those with supplementation of yeast or virginiamycin. The study also indicated the high complexity of the chicken gut microbiome and highlighted the need to document or understand the relation between gut microbial community structure and alternatives to AGP feed additives. The study also indicated that immobilization of EOs in calcium alginate and oil matrix for targeted release of EOs in the chicken gut can be an effective and efficient approach for tackling the menace of AMR. Supplementation of immobilized essential oil microcapsules can easily be adopted in a normal farm setup and hence, may be an effective strategy for tackling the menace of antimicrobial resistance in broiler chicken production.

## Figures and Tables

**Figure 1 microorganisms-11-01960-f001:**
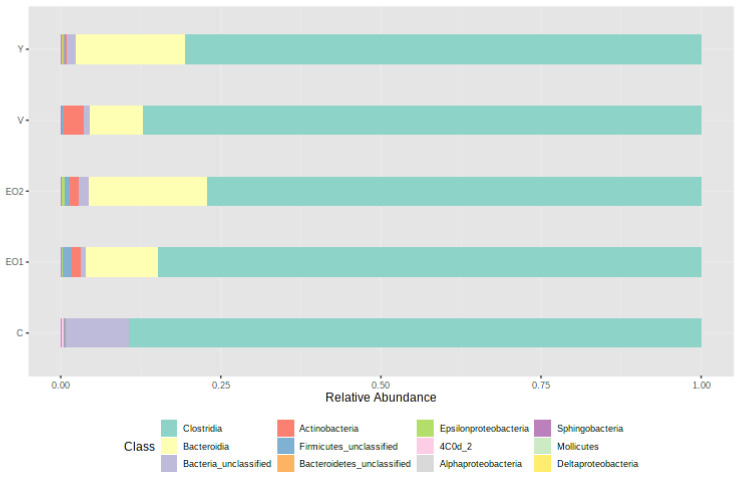
Bar-plots of the average normalized relative abundance of the 12 most abundant bacterial taxa, identified to class level, as found in different groups.

**Figure 2 microorganisms-11-01960-f002:**
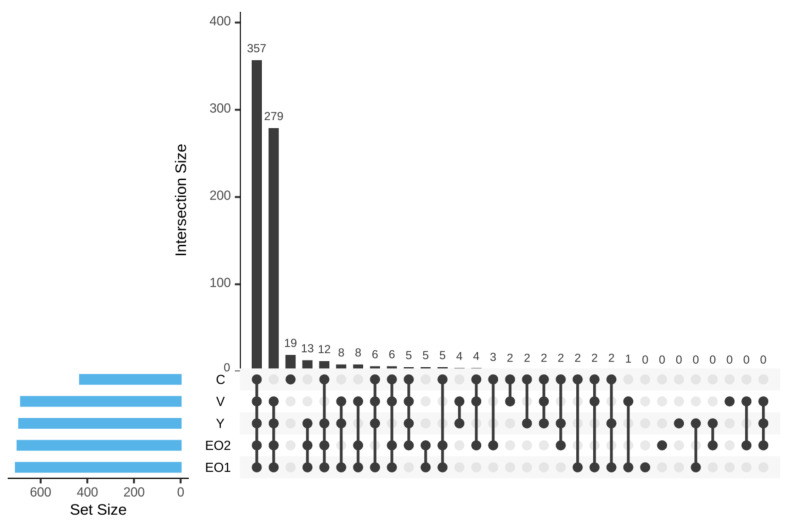
UpSet diagram visualizing intersections of sets of OTUs among different groups.

**Figure 3 microorganisms-11-01960-f003:**
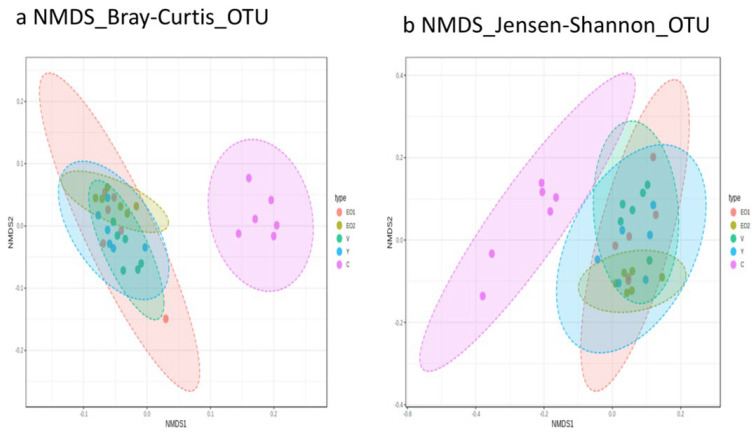
Beta diversity among treatments. Beta diversity plots visualized using nonmetric multidimensional scaling-based ordination at the OTU level for different beta diversity metrics: (**a**) Bray–Curtis index, (**b**) Jensen–Shannon. C, control; V, virginiamycin; Y, yeast; EO1, EO 200 g/t; EO2, EO 400 g/t.

**Figure 4 microorganisms-11-01960-f004:**
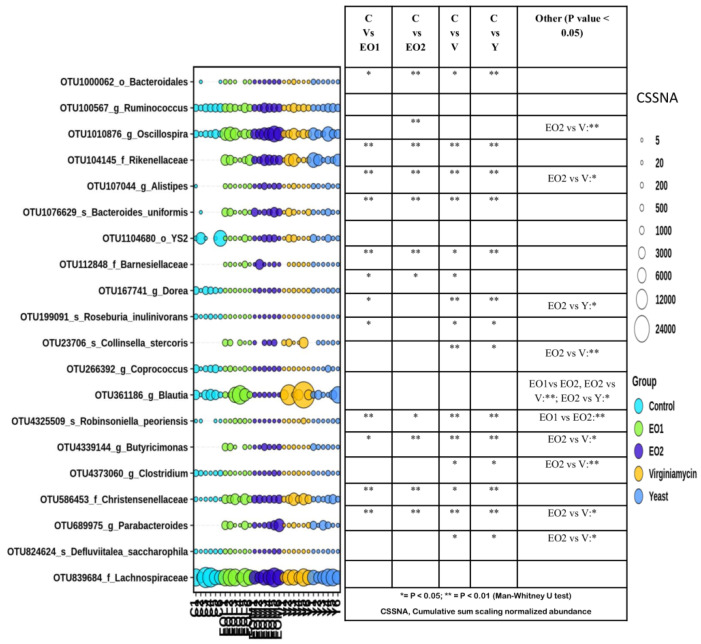
Differential abundance of gut microbiota in different groups at the OTU level. The top 20 OTUs with a significant difference in abundance among groups identified with DESeq2 and passing a false discovery rate filter were plotted. The sizes of the bubbles in the bubble plot indicates the log-transformed (LN(2)) normalized (cumulative sum scaling) abundance of each OTU.* = *p* < 0.05; ** = *p* < 0.01.

**Figure 5 microorganisms-11-01960-f005:**
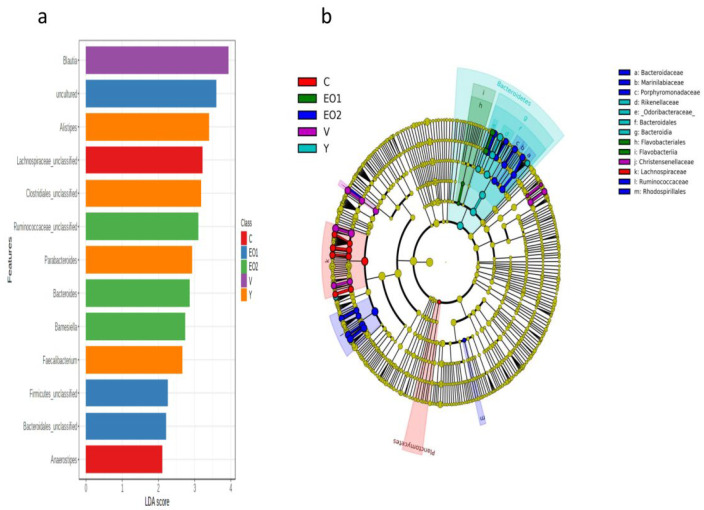
Group-specific biomarkers: (**a**) Genus-level biomarkers, identified using linear discriminant analysis effect size (LEfSe) analysis using the Kruskal–Wallis test (*p* < 0.05) with linear discriminant analysis (LDA) score > 2.0, (**b**) Cladogram representation of differentially abundant microbiota at different taxonomic levels at LDA score > 3.5. The taxonomic levels of the phylum are labeled, while the order and the genus are abbreviated, with the colors indicating the groups with the highest abundance.

**Table 1 microorganisms-11-01960-t001:** Effect of supplementing feed additives on the performance of broiler chicken breeding.

	C	V	Y	EO1	EO2	SEM	N	*p*-Value
Performance								
1–14d								
BWG	329.2	349.8	332.6	349.9	353.6	9.09	12	0.229
FE	1.362c	1.286a	1.373c	1.331c	1.332ab	0.012	12	0.003
1–21d								
BWG	732.9	749.7	743.4	769.9	764.7	15.8	12	0.183
FE	1.389b	1.334a	1.366b	1.392b	1.385b	0.013	12	0.006
1–28d								
BWG	1224	1266	1226	1258	1254	13.4	12	0.150
FE	1.482	1.427	1.455	1.451	1.459	0.018	12	0.256
1–35d								
BWG	1811b	1765a	1806b	1854b	1849b	16.3	12	0.001
FE	1.566	1.562	1.537	1.519	1.548	0.013	12	0.089
1–42d								
BWG	2428ab	2376a	2433ab	2441bc	2492c	21.0	12	0.004
FE	1.652b	1.674b	1.637ab	1.599a	1.597a	0.020	12	0.039

C, control (basal diet); V, virginiamycin (40 g/ton); Y, yeast (S. cerevisiae, 25 billion CFU/g;500 g/ton); EO1, essential oil mix immobilized microspheres (200 g/ton), EO1, essential oil mix immobilized microspheres (200 g/ton); BWG, cumulative body weight gain, g; FE, body weight gain/feed intake; *p*, probability, N, number of replicates; SEM, standard error of the mean. Means having common letters in a row do not vary significantly (*p* < 0.05).

**Table 2 microorganisms-11-01960-t002:** Effect of the feed additive on normalized fold changes of gene expression of intersinal porosity and immunity related genes.

Group	Claudin 1	Claudin 2	Occludin	IFNG	IL6	NOD1	TLR2A
V	2.17a	0.650	2.92	6.24b	2.67a	0.986a	1.639a
C	0.971a	0.970	0.979	0.97a	1.0a	0.981a	1.00a
EO1	95.6c	0.345	1.41	117.5c	182.5b	20.4c	312.6c
EO2	4.81b	0.321	2.67	8.61b	400.1c	2.54b	15.5b
Y	13.1b	0.331	1.14	109.5c	182.5b	5.22b	61.6c
*p* value	0.002	0.160	0.330	<0.0001	<0.0001	<0.005	<0.0001

C, control (basal diet); V; virginiamycin (40 g/ton); Y, yeast (S. cerevisiae, 25 billion CFU/g; 500 g/ton); EO1, essential oil mix immobilized microspheres (200 g/ton); EO2, essential oil mix immobilized microspheres (400 g/ton); IFNG, interferon gamma; IL6, interleukin 6; NOD1, nucleotide binding oligomerization domain 1; TLR2A, toll-like receptor 2A. Means having common letters in a column do not vary significantly (*p* < 0.05).

## Data Availability

The sequence datasets generated in this study have been deposited in the Sequence Read Archive of the NCBI (accession numbers: PRJNA641245 and PRJNA817574).

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
