# Peer review of "An Immobilized Form of a Blend of Essential Oils Improves the Density of Beneficial Bacteria, in Addition to Suppressing Pathogens in the Gut and Also Improves the Performance of Chicken Breeding"

_microorganisms, 2023, doi:10.3390/microorganisms11081960_

Round 1

Reviewer 1 Report

Attached

Author Response

Detail response to comments made by reviewers

Reviewer#1

Reviewer’s comment:Line 59-61 provide reference

Authors’ response: reference has been added in the revised MS (L62)as suggested

Reviewer’s comment:Line 62-65 provide reference

Authors’ response:reference added in the revised MS(L69)

Reviewer’s comment:Authors should provide a reference to their previous study

Authors’ response: reference has been added in the revised MS( L84)as suggested

Reviewer’s comment: The authors mentioned mucosal immunity. Authors should add an expression study for the mucin genes or change mucosal immunity to the epithelial barrier

Authors’ response: changed mucosal immunity to the epithelial barrier (L500) in compliance with the comment

Reviewer 2 Report

Dear Author:

Keywords .- please use only broiler chicken , remove chickens .

Please inform how the feed is keep in order to avoid loss of effectivity of EO . The feed is ofered daily . Please how the present data can be estrapoled to at regular farm.  Please make a coment on the final conclusion . The inmobilized form keep the 80 % of bioactivite , that percentage of effectivity can be find en a regular farming system 

References are good and you manage recent data but in my opinion it would be interesting when you discuss the AGP action why you not mrntion the hypothesis of Niewold ,T. 2007 . Poult.Sci.  86:605-609

Author Response

Detail response to comments made by reviewers

Reviewer#2

Reviewer’s comment: Keywords .- please use only broiler chicken , remove chickens

Authors’ response:necessary changes made in the revised MS (L44) as suggested

Reviewer’s comment: Please inform how the feed is keep in order to avoid loss of effectivity of EO . The feed is ofered daily . Please how the present data can be estrapoled to at regular farm.  Please make a coment on the final conclusion . The inmobilized form keep the 80 % of bioactivite , that percentage of effectivity can be find en a regular farming system 

Authors’ response:Feed was prepared for each phase (15d) and stored in a bin with lid. The experiment was carried out in regular farm so results are applicable in regular farm. Encapsulation itself reduce speed of loss of EO. If we add EO directly in feed most of the EO that are consumed are rapidly aborbed in foregut and thus donot reach site of action ie hindgut but in case of immobilization most of the EO is delivered in hindgut. We have already mentioned this in text. In view of the comments and in compliance with suggestion we have further modified the text of the revised MS (L106-107, 133-134 and 650-652 and added additional references (l69) to clarify these aspects, and added one comment at end of the conclusion.

Reviewer’s comment: References are good and you manage recent data but in my opinion it would be interesting when you discuss the AGP action why you not mrntion the hypothesis of Niewold ,T. 2007 . Poult.Sci.  86:605-609

Authors’ response: We have added the reference and the hypothesis in introduction as suggested (L55-56 and L60).

Reviewer 3 Report

I put some comments in the text.

The paper is very well written, with well-explained results and methods that are up-to-date.

The language is understandable to me and I have no objections to the English language, but a native speaker should look at the text.

Author Response

Reviewer#3

Reviewer’s comment: Corrections suggested on the body of Manuscript

Authors’ response: All the corrections have been incorporated in the revised MS as suggested (kindly see the MS in track change mode)

Note: we have included a revised MS in track change mode for easy identification of corrections.